# Single cell atlas of *Xenoturbella bocki* highlights limited cell-type complexity

Helen E. Robertson[1,2,3,8], Arnau Sebé-Pedrós ®[4,5,6,8], Baptiste Saudemont[2], Yann Loe-Mie ®[2], Anne-C. Zakrzewski ®[3], Xavier Grau-Bové ®[4], Marie-Pierre Mailhe[2], Philipp Schiffer ®[3,7], Maximilian J. Telford ®[3] ✉ & Heather Marlow ®[1,2] ✉

Phylogenetic analyses over the last two decades have united a few small, and previously orphan clades, the nematodermatids, acoels and xenoturbelids, into the phylum Xenacoelomorpha. Some phylogenetic analyses support a sister relationship between Xenacoelomorpha and Ambulacraria (Xenambulacraria), while others suggest that Xenacoelomorpha may be sister to the rest of the Bilateria (Nephrozoa). An understanding of the cell type complements of Xenacoelomorphs is essential to assessing these alternatives as well as to our broader understanding of bilaterian cell type evolution. Employing whole organism single-cell RNA-seq in the marine xenacoelomorph worm *Xenoturbella bocki*, we show that Xenambulacrarian nerve nets share regulatory features and a peptidergic identity with those found in cnidarians and protostomes and more broadly share muscle and gland cell similarities with other metazoans. Taken together, these data are consistent with broad homologies of animal gland, muscle, and neurons as well as more specific affinities between *Xenoturbella* and acoel gut and epidermal tissues, consistent with the monophyly of Xenacoelomorpha.

*X enoturbella bocki* is a marine worm with long-contested phylogenetic affinities. Initially described as a turbellarian flatworm, *X. bocki* was first described by Westblad in 1949 from specimens collected from the muddy bottom of the Gullmarsfjord on the West coast of Sweden[1]. *X. bocki* is a ~2 cm long yellowy-orange or brown slightly flattened worm (Fig. 1A). It has a blind-ended gut, no clear nervous condensations (nerve chords or a brain), coelomic cavities, discrete gonads or filtratory organs[1–3]. Five species have been described within the genus *Xenoturbella* in the past few years, all of which have a very similar morphology to *X. bocki*, differing most obviously in their colours and size with the largest, *X. monstrosa*, being as much as 20 cm long[4].

Xenoturbellida (the higher-level group that contains *Xenoturbella*) is widely accepted to be the sister group of a second group of morphologically simple, free living marine worms—the Acoelomorpha, which together form the phylum Xenacoelomorpha. The Xenacoelomorpha are no longer considered to be close to the Platyhelminthes but their true affinities have been highly contentious[5–7]. While a number of phylogenetic studies placed Xenacoelomorpha as the sister group of all other Bilateria (the Nephrozoa), recent work suggests this tree is influenced by systematic error and the true affinity of Xenacoelomorpha is as the sister group of the Ambulacraria (Hemichordata and Echinodermata) to form the group Xenambulacraria[8].

[1]Department of Organismal Biology and Anatomy, The University of Chicago, Chicago, IL, USA. [2](Epi)genomics of Animal Development Unit, Department of Developmental and Stem Cell Biology, Institut Pasteur, Paris, France. [3]Centre for Life's Origins and Evolution, Department of Genetics, Evolution and Environment, University College London, London, UK. [4]Centre for Genomic Regulation (CRG), Barcelona Institute of Science and Technology (BIST), Barcelona, Spain. [5]Universitat Pompeu Fabra (UPF), Barcelona, Spain. [6]ICREA, Barcelona, Spain. [7]Institute of Zoology, Section Developmental Biology, University of Cologne, Köln, Wormlab, Germany. [8]These authors contributed equally: Helen E. Robertson, Arnau Sebé-Pedrós. ✉e-mail: m.telford@ucl.ac.uk; hmarlow@uchicago.edu

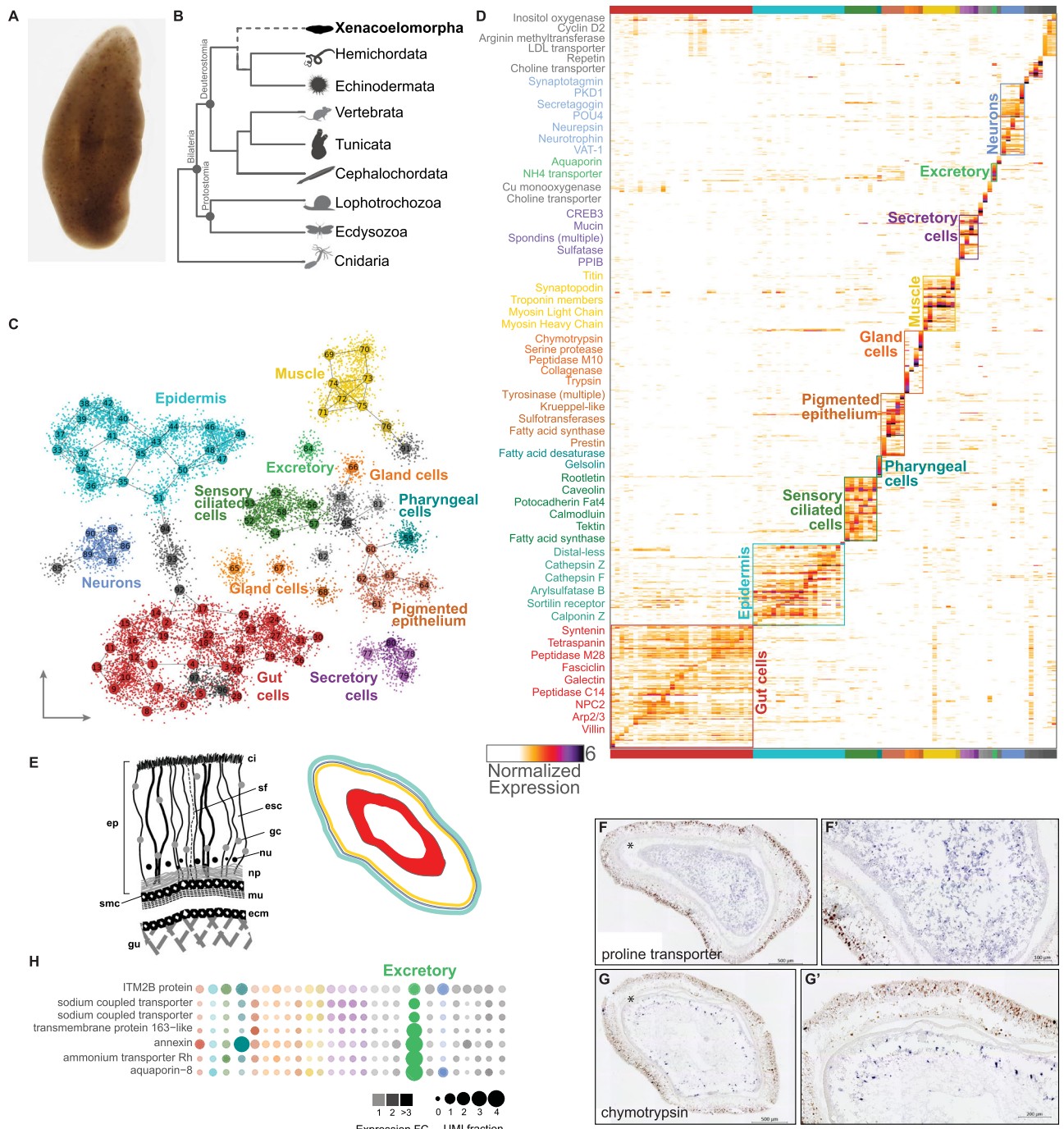

**Fig. 1 | *Xenoturbella* single-cell RNA-seq identifies spatially distinct cell types and a minimally complex neural cell type complement. A** A light micrograph of a live *Xenoturbella bocki*. **B** Phylogenetic tree showing the recently proposed position of *X. bocki* as sister to Ambulacraria. **C** 2D projection of 97 metacells (indicated with numbers) and 12,350 *X. bocki* cells. **D** A cell by gene heatmap of cells from **C** showing expression similarity within the distinct tissue types. **E** A schematized diagram of a cross-section of *Xenoturbella* tissues: ci (cilia), sf (supporting filament), esc (epidermal supporting cell), gc (gland cell), nu (nucleus), np (nerve plexus), m

(muscle), ecm (extracellular matrix), gu (gut), smc (sub-epidermal membrane complex), ep (epidermis). Right, schema of a horizontal cross-section of a *Xeno-turbella* individual in which endodermal (red), ectodermal (aqua), and mesodermal (yellow) derivatives are indicated. **F**, **G** In situ hybridization of cell-type markers identified from the scRNA-seq clustering. **F**, **G** are high magnification images of **F'**, **G'**, respectively. **H** Markers associated with excretion co-localize in an unknown cell type (green). Images used to generate 1b were obtained from phylopic.org.

Whatever their true phylogenetic placement (whether as sister group of Nephrozoa or of Ambulacraria) (Fig. 1B), the morphological features of Xenoturbellids are critical to our understanding of animal body plan evolution[6,9]. Two major branches within animals constitute the Bilateria, a superphyletic assemblage of bilaterally symmetrical clades, the protostomes and the deuterostomes[10]. Within

deuterostomes, the diversity of body plans and nervous system organizations have presented a challenge for the reconstruction of ancestral deuterostome characters. Recent evidence suggests that profound differences between ambulacrarian and chordate body plans could even be explained by paraphyly of the deuterostome clades[8]. It follows that the position of Xenacoelomorphs as sister of Ambulacraria, makes

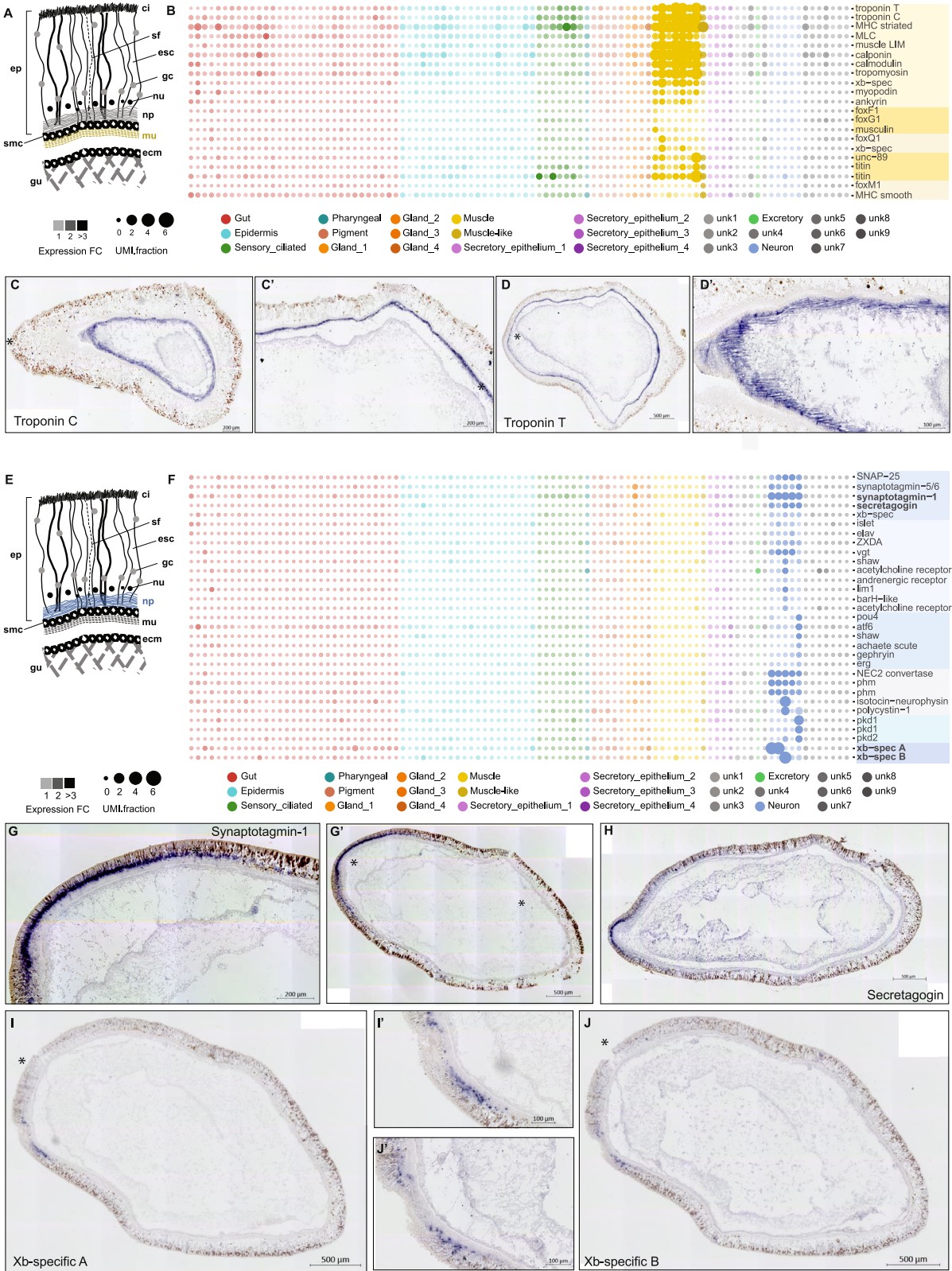

**Fig. 2 | Muscle and neural metacell markers localize to histologically distinct muscle cell types and cell bodies in the subepithelial nerve plexus, respectively. A** Schematic indicating the position of muscle in *Xenoturbella*. **B** Muscle markers are enriched in a group of eight metacells (yellow). **C** In situ hybridization of troponin transcripts localization to muscle. **C'** High magnification image of striations in stained tissue from **C**. **D** Troponin staining in muscle. **E** Schematic indicating the position of the subepidermal membrane complex (smc). **F** A group of

metacells (blue) expressing transcripts encoding channel proteins, neuropeptide processing proteins, secretory proteins and transcription factors.
**G**, **H** Synaptotagmin I **G**, **G'** and secretagogin **H** expression are localized to the anterior nerve plexus. **G** is a high magnification image of **G'**. **I**, **J** *Xenoturbella* specific transcripts are localized to the nerve plexus. **I'** and **J'** are high magnification images of **I**, **J**.

them increasingly relevant for our understanding of non-protostome body plan evolution. If deuterostomes really are a paraphyletic clade, the Xenacoelopmorph body plan may serve as a proxy for a simple Xenambulacrarian ancestor. As such, investigations of *Xenoturbella*'s cellular diversity and transcriptional programs are of particular interest especially in reference to excretory organs and nervous system cellular diversity.

Our current understanding of the morphology of *Xenoturbella* relies, for the most part, on classical morphological and histological descriptions. *Xenoturbella*'s most notable external features are a ventrally placed mouth acting as the single opening to a large central blind gut lumen, a circumferential ring furrow dividing front and rear of the body, and lateral furrows at the anterior end. Internally there is an anterior statocyst, a thick, ciliated epidermis with a basiepithelial nervous system and layers of muscles surrounding the gut. Our knowledge of the cell type diversity of *Xenoturbella bocki* at the level of either cellular morphology or gene expression is minimal.

Here, we use whole-organism single cell RNA sequencing (scRNA-seq) as a high-throughput approach to characterise the cell type expression programs in adult specimens of *Xenoturbella*. In order to validate our findings and to link the cell states we have identified to the locations and morphology of cells in the animal, we have developed an in situ hybridisation (ISH) protocol that works on sectioned animals.

## Results

### Complement of cell states in *Xenoturbella*

We sampled single cell transcriptomes from four adult specimens of *Xenoturbella bocki* collected from the Gullmarsfjord, Sweden. We used the MARS-seq, microplate-based single-cell sequencing protocol[11] (for details see Methods). We sequenced single-cell libraries to an average depth of 33,870 reads per cell (Table S1). The average total mappability to the *Xenoturbella* genome was 78% of reads, with 26% uniquely mapped reads. 12,350 cells with more than 100 transcripts per cell (determined by Unique Molecular Identifiers, UMIs) were retained for downstream analyses. The median number of transcripts measured per retained cell was 418 and we detected expression for a total of 11,304 genes (> = 10 total UMIs) (Fig. S1A, B)

We applied the Metacell algorithm[12] to group cells into transcriptionally coherent clusters (metacells) (Fig. 1c, S1C–E). Each metacell contained between 41 and 291 single-cells. By examining a graph-based 2D projection (Fig. 1C) and gene co-expression patterns (Fig. 1D, S2), we identified 26 distinct cell type/tissue-level groups of metacells characterised by specific transcriptional signatures. These metacell groups include putative muscles, neurons, epithelial and digestive gland cells, amongst others (Fig. 1C, D). Gut cells (30%) and epidermal cells (22%) are the most abundant cell types in *Xenoturbella*, followed by muscle (8%), ciliated cells (7%), neurons (5%) and pigmented cells (5%) (Fig. 1C).

To localise the populations of cells characterised by each of the cell states identified by their transcriptomic profile, we identified marker genes whose expression was highly specific for each metacell population. We characterised the expression of a subset of these markers in preserved tissue sections of adult *Xenoturbella* using in situ hybridization (ISH). This allowed us to associate the cell clusters we had identified with the corresponding cells in the adult animal providing information on the relative positions of different cell types and their morphology, size, and abundance (Fig. 1E).

### Gut and digestive cells

The blind-ended *Xenoturbella* gut constitutes a large central tissue with a lumen that is clearly identifiable in horizontal-sections (Fig. 1E)[2]. Reflecting the large proportion of the animal that is made up of gut tissue, we found that the largest group of metacells, defined principally by the expression of a number of transporter proteins, is identified as the gut tissue by ISH (Fig. 1F). We have identified four additional

groups of cells that express digestive enzymes and proteases, and which are likely to belong to a digestive gland population. One of these secretory gland cell populations is characterised by expression of chymotrypsin, and ISH locates this cell type to the gastrodermis of *Xenoturbella* (Fig. 1G).

We also found a population of cells that expresses several genes—sodium transporters, an Rh-type ammonium transporter, and an ortholog of aquaporin-8 (Fig. 1H). These aquaporin and ammonium transporter genes are enriched within this cell population, distinct from the gut metacell. Cross-species comparison found no similarity between this *Xentourbella* metacell and the digestive cell populations of the acoels *Hofstenia miamia* and *Isodiamtera pulchra* (Fig. S4). Interestingly, we found strong similarity between this metacell and the anus of the larva of the sea urchin *Strongylocentrotus purpuratus*, driven by the common expression of cell membrane and cytoskeletal transcripts and *aquaporin-8*.

### Muscle cell types

Three discrete populations of muscles have been described in *Xenoturbella* with different orientations and relative positions—an inner layer of longitudinal muscle adjacent to the gut, a layer of circular muscle surrounding these, and radial muscles connecting the circular and longitudinal muscle layers to the epidermal layer (Fig. 2A). All of these have been described as smooth muscle[13].

Metacells representing muscle cells in *Xenoturbella* were identified by the presence of conserved marker genes including *troponin* and *myosin* homologues. Most muscle cluster cells express *calponin*, *calmodulin*, *muscle LIM* and *myosin light chain*. A subset of metacells expresses the transcription factors *musculin* (*MyoR*), *FoxF1* and *FoxG* and an additional group is specifically enriched for titin-like contractile elements (Fig. 2B).

All eight muscle metacells identified express a striated-type *myosin heavy chain* (*MHC*) ortholog as well as three *troponin* orthologs (*troponin I, C and T*); only *troponin I* and *T* are present in echinoderms and hemichordates. One of these eight clusters of cells (# 76) expressing muscle-effector genes specifically expresses a "smooth" muscle myosin type. These cells, as with the other muscle clusters, co-express other muscle-related transcripts including *troponin* members and *titin*-related genes (Fig. 2B). This cluster also expresses a complement of transcription factors, including zinc finger TFs and a sox TF, which is largely distinct from the the other seven clusters of metacells, which express the homeodomain TF *six1* and the bHLH TF *heart and neural-crest derived* (*HAND*)(Fig. S2). Examining the expression of a number of cell cycle marker genes, we identify cluster #76 to be a likely muscle progenitor population.

To confirm that the clusters identified by muscle-specific genes in our scRNA-seq analysis (Fig. 2B) correspond to the muscles previously characterised by histological analyses, we performed ISH with *XbTroponin-C* and *XbTroponin-T* which are expressed across all the putative *Xenoturbella* muscle clusters and found expression in the prominent muscle layer adjacent to the gut (Fig. 2C, D).

### Neuronal cell types

The nerve plexus of *Xenoturbella* has been described as a simple basiepithelial nerve net, lacking discrete ganglia or nerve cords[3,14]. The location, density and possible identity of neurons has been characterised in a limited number of ultrastructral studies[3,14–16], but there has been no molecular characterization of the diversity of cell types that might be present or their distribution. Both major lineages of the sister clade Ambulacraria—hemichordates and echinoderms—show some condensation of nerve cords[17,18], and different degrees of condensation is seen in acoels[9,19]. It remains unclear whether the 'nerve-net' like organization observed in *Xenoturbella* is an ancestral or derived condition. We aimed to identify neuronal populations among *Xenoturbella* metacells, to identify their putative functional

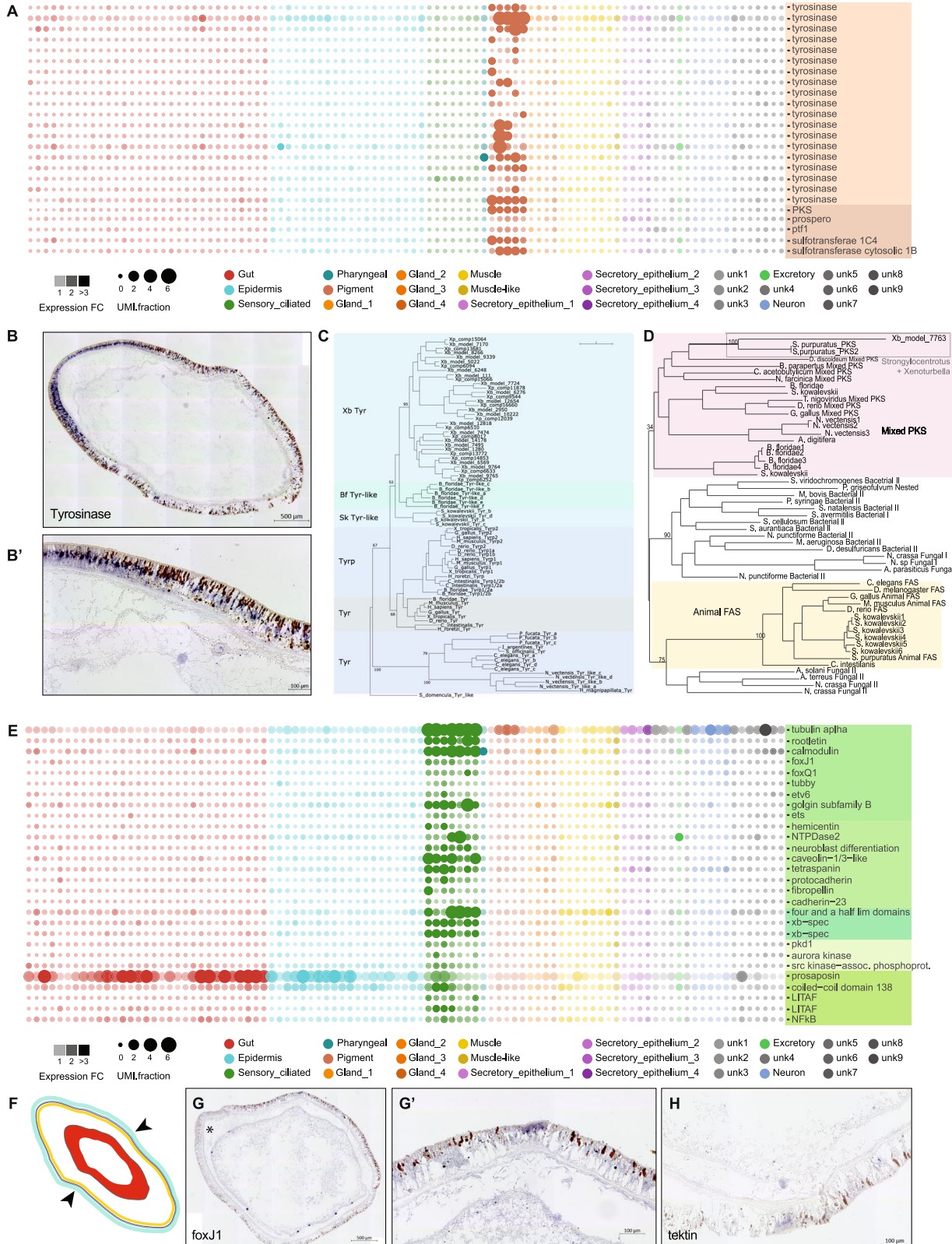

**Fig. 3 | Hallmarks of *Xenoturbella*, the pigmented epidermal cells and the sensory, ciliated transverse band, are associated with metacell groups by in situ hybridization. A** An expansive repertoire of tyrosinases as well as polyketide synthase (pks) are enriched in pigmented epithelial cells. **B** Tyrosinase transcript expression in pigmented epithelial cells. **B'** Is a high magnification image of **B**. **C** Maximum likelihood tree indicating a large, species-specific, expansion of tyrosinase transcripts in *Xenoturbella* and **D** the position of *Xenoturbella* PKS as nested within animal PKS genes. **E** Transverse sensory band enriched markers. **F** Schematic of the position of the transverse sensory band on an intact animal. Expression of the cliiary sensory band markers foxJ1 **G** and tektin **H**. **G'** High magnification image of the region indicated in **G**.

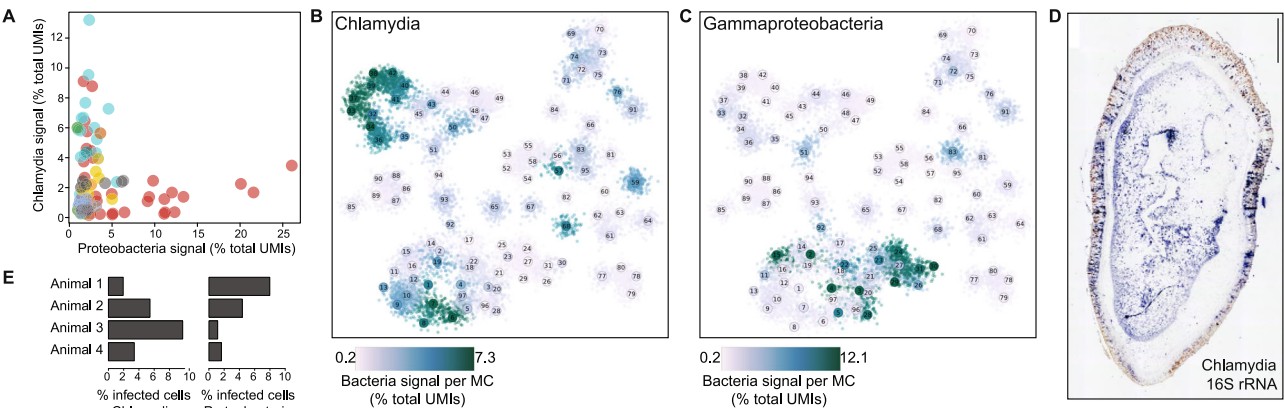

**Fig. 4 | Intracellular microbial populations of *Chlamydia* and Gammaproteobacteria are associated with gut and epidermal tissues. A** Abundance of each microbial population within *Xenoturbella* cells. The proportion of *Chlamydia* vs. Gammaproteobacteria show in the four sampled individuals. **B** *Chlamydia* are associated with gut and epidermal cells. **C** Gammaproteobacteria are associated with gut populations only. **D** An in situ hybridization of 16 s rRNA for *Chlamydia* expressed in both gut and epidermal populations of cells. The scale bar is 500 μM. **E** Bar plots showing the percentage of *Chlamydia* and *Proteobacteria*-bearing cells in each animal surveyed.

identities and to examine their expression of transcriptional regulators.

Histological studies of *Xenoturbella* cross sections have shown that the neuronal cell bodies lie in the region between the bulk of the nerve plexus and the base of the epidermis (Anne Zakrzewski, personal observation) (Fig. 2E). Presumptive neuronal metacells were identified by expression of the pan-neural gene *synaptotagmin* (Fig. 2F). ISH using *synaptotagmin* labelled cells in the basal portion of the epidermis −in the presumptive nerve plexus−with higher levels of expression at the anterior end of the animal (Fig. 2G). These neuronal metacells also expressed markers characteristic of vesicle secretion and of the pre-synaptic synapse, notably, *secretagogin* and *SNAP25* (Fig. 2F). *Secreta-gogin* is localised to the nerve plexus at the very anterior of the animal (Fig. 2H). In addition to these synaptic vesicle genes, enzymes required for neuropeptide processing, indicative of peptidergic neurons (*pro-hormone convertase* and *peptidylglycine alpha-hydroxylating mono-oxygenase* were expressed across all neuron subtypes) (Fig. 2F). We also identified two broad groups of neurons expressing either the transcription factors *POU4* or *islet* (Fig. 2F). Neurons marked by *POU4* are likely to be sensory neurons expressing members of the *erg* family of K+ channel genes as well as a diversity of PKD channels. The neural bHLH member *achaete-scute* is also specifically upregulated in *POU4* positive neurons. A broader subset of neurons is identified by the expression of *islet*. We found *islet* to be co-expressed with the gluta-mate transporter *vgt*, and with *elav* and *irx*. A specific group of *islet* positive cells is also enriched for *lim1*, and adrenergic and acetylcho-line receptors (Fig. 2F).

We identified several genes expressed in specific neuronal cell clusters for which we could identify no orthologs outside of *Xeno-turbella*. Dot plot mappings of these show expression in distinct neural metacells (Fig. 2F). For one of these orphan genes, ISHs localised expression to a small and specific subset of cells within the anterior nerve plexus (Fig. 2I). A second orphan gene had a more diffuse expression across a defined anterior region of the animal's nervous system (Fig. 2J).

### Pigmented cells
One group of several related metacells was strongly enriched for *tyr-osinase*-family genes (Fig. 3A). Tyrosinases are found across the animal, plant and fungal kingdoms, where they act as the primary rate-limiting enzyme during melanin biosynthesis[20]. In vertebrates, melanin is found in pigment cells; in invertebrates, melanin has been reported to

function in several different roles including wound healing, immune response, and photoreception[21,22].

ISH for one of the *Xenoturbella* tyrosinase members localised expression across the epidermis, with higher expression at the anterior of the animal. This expression in the epidermis is consistent with the Tyrosinase expressing cells corresponding to the subset of pigmented epidermal cells (Fig. 3B).

Tyrosinase proteins and the related tyrosinase-like proteins show independent lineage-specific expansions in a number of metazoan groups[20,23,24]. For example, tyrosinase-related proteins (tyrp), which are identifiable by conserved residue changes in their metal binding domains, are uniquely found in chordates[20]. A phylogenetic analysis of tyrosinase and tyrosinase-related sequences show that *Xenoturbella* tyrosinase sequences represent a *Xenoturbella*-specific expansion (Fig. 3C). Our tree of Metazoan tyrosinase related proteins groups *Saccoglossus*, vertebrate and *Branchiostoma* tyrosinases and tyrosinase-related proteins, and places the *Xenoturbella* tyrosinase-like sequences as a sister group to all other deuterostome tyrosinase family proteins.

Cells in the *Xenoturbella* tyrosinase-enriched metacells are also enriched for transcripts of *polyketide synthase-like* (*PKS*). PKSs and the related fatty acid synthases (FASs) are a diverse family, found across the eubacteria and eukaryotes[25]. In a phylogeny of PKS and FAS sequences from across eubacteria and eukaryotes, the XbPKS1 sequence groups with high confidence with sequences (SpPKS) from an echinoderm (*Strongylocentrotus purpuratus*) (Fig. 3D).

In *S. purpuratus, SP-PKS1* is necessary for the biosynthesis of echinochrome−one of the major pigment classes present in echinoderms[26]. Several other genes, including sulfotransferases, are necessary for pigment synthesis[26]. Both *PKS* and *sulfotransferases* are co-expressed with the *tyrosinases*, indicating some similarity to pig-ment synthesis. However, cross-species comparisons between cell types in *Xenoturbella* and sea urchin revealed no specific relationship between the sea urchin pigment cells and cells expressing tyrosinase in sea urchin (Supplementary Fig. 4). No tyrosinase or tyrosinase-like members have been identified in the echinoderms[24]. All pigment cell clusters found in *Xenoturbella* express *dopt* and *sulfotransferase*, but we did not detect enrichment of any *fmo* members or *gcm*.

### Ciliary furrow
We identified seven metacells marked by the expression of numerous ciliary-related genes - including *tubulin, calmodulin, rootletin, FoxJ1*

and *tubby* (Fig. 3E). In situ hybridisation for *FoxJ1* localised expression to two opposing regions midway along the anteroposterior axis, which we identify as the ciliated circumferential furrow (Fig. 3G) midway along the anteroposterior axis of *Xenoturbella* (Fig. 3F). The function of the circumferential ciliary furrow in *Xenoturbella* is not known. ISH for an ortholog of the tektin gene *Tekt4*, necessary for microtubule stabilization[27], also localized expression to this circumferential furrow (Fig. 3H). ISH for a *Xenoturbella* orphan gene specific to the ciliary metacell group, showed expression in both the ciliary furrow and the anterior nerve plexus (Fig. S3). We did not find expression by ISH of any other ciliary marker gene in the neuronal metacells, nor neuronal marker genes in the ciliary metacells.

We looked further into the genes expressed in the ciliary metacluster and found many markers associated with extracellular matrix (ECM) remodelling and maintenance, such as *hemicentin-1, ets* factors*, cadherin* members including *cadherin-23, collagen, integrin* and *fibropellin*, and a number of genes necessary for conveying information between the ciliary membrane and ECM, including *PKD1, AURKA, SRC*. Interestingly, we also find genes implicated in bacterial response, including *Saposin, LITAF*, and NFkB (Fig. 3E).

### Cell type-specific endosymbiotic bacteria

*Xenoturbella* has previously been shown to harbour two types of bacteria in the gastrodermis; one related to *Chlamydia* and the other to Gammaproteobacteria[28,29]. We identified genes showing similarity to sequences from both *Chlamydia* and Gammaproteobacteria in our *Xenoturbella* scRNA-seq data supporting an intracellular endosymbiotic relationship. By measuring bacterial transcript abundances across metacells, we show that the *Chlamydia* is found both in the gastrodermis and the epidermis, while the Gammaproteobacterial signal is found primarily in gut metacells (Fig. 4A). We found that the two different bacteria infect different subsets of cell populations within the gastrodermis, with a small population of cells that are infected by both types of bacteria (Fig. 4B, C). We found that *Chlamydia* also infect a group of cells within the epidermal population (Fig. 4A, B). Confirming this predicted pattern, In situ hybridisation against the *Chlamydia 16 S* rRNA localised this bacterium to both the gut and epidermis of the animal (Fig. 4D).

Interestingly, the percentage of cells we detect which are infected by either endosymbiont varies across the animals sampled (Fig. 4A). However, the percentage of cells infected by either endosymbiont is inconsistent across the animals sampled. In three out of four of the animals sampled ~10% of cells contained bacteria; the fourth replicate had just ~5%. The relative contribution of each bacterial type to total infection is also variable between animals (Fig. 4E). Discrepancies between frequency and type of bacterium did not obviously correlate with the time of year at which the different animals were collected although the number of samples was limited.

### Unknown *X. bocki* cell types

In our characterization and classification of *X. bocki* cell types, we identified nine clusters which were difficult to assign to a particular identity based on the expression of specific marker genes. To better identify affinities with the cells known from other Xenacoelomorph lineages, we used a cross-species analysis[30]. These "unknown" lineages showed limited similarity to sea urchin cell types[31] including those of the coelomic pouches, ciliary band, and oral ectoderm (Supplemental Fig. 4). However, when comparisons were made to lineages in which adult pluripotent stem cells (aPSCs) have been identified, namely the acoelomorphs *H. miamia* and *I. pulchra* as well as the cnidarian *N. vectensis*[32–34], we identified similarity between unknown cluster 9 and aPSCs or progenitors (Supplementary Fig. 4). Upon further inspection, much of this similarity is driven by the expression of histones and ribosomal proteins. However, shared expression of transcripts of

RRMs, elongation factors and transcripts for *polyA-binding protein* (*PABP*) were also identified.

## Discussion

Here we report the use of whole-organism scRNA-seq in conjunction with *ISH* to characterise the cell type repertoire of *Xenoturbella bocki*. This single cell analysis provides a comprehensive molecular profiling of cell states in this animal, and the parallel use of ISH for highly specific and enriched genes provides a spatial context for the cell types we identified. While the broad categorisation of cell types we find is broadly consistent with prior morphological descriptions of the animal, the molecular fingerprints of these cell types allow us to identify potentially homologous cell types and give unique insights into the biology and life history of the animal.

The proportions of different cell types represented in our data reflect the abundance of these cell types in the animal: the gut and epidermis, which are the largest tissue layers seen in histological analysis, comprise over half of the metacells in our clustered dataset. Although we were able to assign an identity to the majority of metacells identified in our analysis, some metacells, representing between 0.8 and 1.8% of the sampled cells in our dataset, remain unidentified.

We find populations of digestive gland cells in *X. bocki* that do not express transporters, indicating that *Xenoturbella* has distinct populations of digestive gland cells that are separate from the nutrient transporting and absorptive cell populations of its blind gut. We also find a distinct population of cells expressing *aquaporin-8* and an Rh ammonium transporter, which have very little similarity to the digestive cell populations of the acoels *H. miamia* or *I. pulchra*. In *I. pulchra*, an aquaporin member, the bilaterian ultrafiltratory gene nephrin/kirre and transporters were found to be co-expressed in a broad population of cell types[35], indicative of a putative excretory function. The distinct identity of this *Xenoturbella* metacell could be indicative of a discrete osmoregulatory role, separate from that of the digestive gland and gut cells, and different to the condition found in acoels. However, we have not been able to produce ISH data that allow us to identify these cells.

We find *Xenoturbella* neurons to be significantly less diverse than those characterized in acoelomorphs[32], another of the Xenacoelomorph lineages. Interestingly, *Xenoturbella* neurons express hallmarks of metazoan neurons including *islet-1, brn3/pouIV* and the proneural bHLH transcription factor *achaete-scute*, indicating that they share some features of an ancestral neural regulatory program. PouIV has been identified as serving as a neural identity factor in lineages across Bilateria including flies and mice and was recently identified to be functionally required for the development of neurons of the sea anemone *Nematostella vectensis*[36–38], suggesting a role in the earliest nervous systems, those found in lineages which predate Bilateria. Similarly, the lim-homeodomain gene *islet-1* has been identified in cnidarian neurons[39] and in neurons and neuroendocrine cells in the Bilateria[40,41].

Expression of the pre-synaptic complex components required for vesicle secretion as well as the neuropeptide processing pathway members in these cells strongly suggests that they function as peptidergic neurons. Recent comparative work in invertebrate nervous systems suggests that neuropeptidergic control of simple behavioural circuits as well as paracrine signalling of neuroendocrine-like cells was a feature of early nervous systems[42–44]. These features have been preserved in marine invertebrate nervous systems broadly supporting a "chemical brain hypothesis" for nervous system evolution[45]. The conservation of this organization in *Xenoturbella* strongly suggests that this simple peptidergic nerve plexus represents an ancestral ground plan for animals that has been preserved in the Xenacoelomorph lineage.

The *Xenoturbella* muscle metacells have the molecular hallmarks of striated muscle despite being morphologically smooth. Similar

observations have been made regarding acoel musculature, and suggests that this feature could represent a synapomorphy of Xenacoelomorpha. This finding provides some support for the independent evolution of striated muscle in separate lineages by the repurposing of core bilaterian-specific proteins.

Recent phylogenetic studies place Xenocelomorpha (Xenoturbellids and Acoels) as the sister clade to the Ambulacraria[7–9]—a lineage which includes the hemichordates (enteropneusts and pterobranchs) and echinoderms—despite few putative morphological synapomorphies of the Xenambulacraria (Xenacoelomorpha plus Ambulacraria clade). While Xenacoelomorpha lack characters such as gill slits, a coleom and a through gut that are likely to have existed in the Ambulacrarian ancestor, they share some characters with possible affinities to ambulacrarian lineages. Among these are the pigment cell population, and the cells of the transverse ciliary band.

In the larval sea urchin, pigment cells develop from primary mesenchyme, a mesodermal cell type. These cells become pigmented by "echinochrome" generated by the PKS and sulfotransferase gene products. Similarly, we find that *Xenoturbella* pigmented cells express both PKS and sulfotransferase as well as a massively expanded repertoire of tyrosine hydroxylase genes. Previous work has not identified tyrosinases in sea urchins. This suggests, that while some enzymes may be shared for the production of pigmentation in urchins, the expansion of *Xenoturbella* tyrosinases is a lineage specific event and may underlie the elaboration of a lineage-specific cell type.

Ciliary bands are present on both the larvae of echinoderms and on the tornaria larvae of hemichordates, where they are necessary for locomotion and feeding[46,47]. In hemichordates, ciliary bands persist through metamorphosis into the adult. The molecular fingerprint of the *Xenoturbella* ciliary cell types is unlike that found in the hemichordates, echinoderms, or that reported in the acoel *Hofstenia*. The primary ciliary markers, *foxj1* and *tubby*, suggest that the circumferential ciliary furrow in *Xenoturbella* may be important for integrating environmental cues, such as fluid-flow or external compression, but we do not find the expression of genes that are otherwise expressed in the neural cluster, or any marker genes that have been found in the neural ciliary band of larval echinoderms. ISH for one *Xenoturbella*-specific gene, with no orthologs in other taxa, showed expression in the ciliary furrow and anterior nerve plexus, but ISH for other ciliary marker genes (including Tubby and Tektin) showed no expression outside of the ciliary furrow. This may indicate that the ciliary band in *Xenoturbella* is a lineage-specific innovation, unrelated to those found in other ambulacrarians.

## Methods

### Experimental model and subject details

Adult *Xenoturbella bocki* were collected from the Gullmarsfjorden on the west coast of Sweden. A Warén dredge aboard the Oscar von Sydow research vessel (Kristineberg Center for Marine Research and Innovation, Gothenburg University) was used to collect soft mud at a depth of ~60–100 m. Mud was subsequently filtered through two sieves of decreasing pore sizes, so that larger organisms were removed first, and *Xenoturbella* (of approximately 5–20 mm) were retained on the mesh of the second sieve with all mud washed away. The contents of the second sieve were washed into large shallow trays, and *Xenoturbella* specimens were identified by eye. Animals were kept unfed in sealed falcon tubes filled with deep-sea fjord water at 4 °C.

Four adult specimens were sampled at different times for FACS sorting. Animals were dissociated by placing them in full strength calcium/magnesium-free and EDTA-free artificial sea-water (CMFSW). Animals were then transferred to 300 μL CMFSW supplemented with LiberaseTM (Roche, #05401119001) at a concentration of 50 μg/ml. Dissociation was carried out at room temperature by pipetting using gelatine-coated pipette tips of decreasing diameter to disrupt the cell suspension over a period of 15 min. Dissociation was stopped by the addition of one-tenth volume of 500 mM EDTA solution to a final concentration of 50 mM. 3 μg Calcein AM (Thermo #C3100MP) and 2.25 μg Propidium Iodide (PI) (Thermo #P3566) were added to the cell suspension to identify live cells. Cells were then placed on ice and not washed out of the Calcein/PI solution before sorting.

Dissociated cells were sorted using a BD FACSAria III fluorescently-activated cell sorting machine into 384-well capture plates with each well containing 2 μl of cell lysis solution. Live cells were selected by sorting Calcein positive/PI negative cells. Doublet/multiplet exclusion was performed using FSC-W versus FSC-H. Lysis solution contained 0.2% Triton X-100, RNaseIN and barcoded poly(T) reverse-transcription (RT) primers for single cell RNA-seq. Immediately after cell distribution, plates were briefly spun down to ensure complete immersion of cells in the lysis solution and stored at −80 °C. Four empty wells were left in each plate as a control.

### Single-cell RNA-seq

Single cell libraries were prepared according to the MARS-seq method of plate-based single-cell transcriptomic sequencing as previously described in detail[11]. In brief, 384-well plates of sorted cells first undergo reverse-transcription in individual wells where cell-specific barcodes and a unique molecular identifier (UMI) are added and are subsequently pooled in plate-specific libraries for second-strand synthesis and amplification where additional plate-based barcodes are added. All 42 plates (a total of 15,960 single cell libraries) were prepared using the same conditions and reagents. Using a Bravo automated liquid handling platform (Agilent), mRNA was reverse transcribed into cDNA using the lysis oligonucleotide containing unique molecular identifiers (UMIs) and cell barcodes. Residual, unused oligonucleotides were removed by treatment with Exonuclease I (NEB). cDNAs were pooled and linearly amplified using T7 in vitro transcription. The resulting RNA was fragmented and ligated to an oligo containing the pool barcode and Illumina sequences, using T4 ssDNA:RNA ligase. RNA was reverse transcribed into DNA and PCR-amplified with 17 cycles. The size distribution and concentration of DNA in the libraries were measured using Tapestation (Agilent) and Qubit (Invitrogen). scRNA-seq libraries were pooled at equimolar concentrations and sequenced as a paired end run on an Illumina NextSeq 500 with a 75 cycle v2 kit. Read one was 59 bp (covering the pool-specific barcode and cDNA) and read two was 17 bp (covering the well-specific barcode and UMI). We obtained a total of 1086 M reads, with an average depth of 30,330 reads per cell and 19.73 reads/UMI on average.

### MARS-seq read processing and filtering

Reads were mapped onto the *Xenoturbella* genome (Phlipp Schiffer, unpublished data) using bowtie2 (with parameters: *-D 200 -R 3 -N 1 -L 20 -i S,1,0.50 -gbar*) and assigned to gene intervals associated with a custom gene annotation. We extended gene intervals up to 1 kb downstream or until the next gene on the same DNA strand is found. This accounts for the poor 3'UTR annotation, which causes many of the MARS-seq (a 3' biased RNAseq method) reads to map outside genes. Additionally, in order to account for putative unannotated genes, we defined 2000bp bins (not covered by our gene intervals) genome-wide. We retained 592 such regions with >=100 uniquely mapping reads and used them in the cell clustering process.

Mapped reads were further processed and filtered as previously described[48]. In brief, UMI filtering includes two components, one eliminating spurious UMIs resulting from synthesis and sequencing errors, and the other eliminating artifacts involving implausible IVT product distributions that are likely a consequence of second strand synthesis or IVT errors. The minimum FDR q-value required for filtering was 0.2.

## Metacell and clustering analysis

We used the MetaCell package[12] to select gene features, to construct gene modules and to create projected visualization of the data, using parameters as described below.

We applied preliminary cell filtering based on total UMI counts using a permissive threshold of 100 UMIs. For gene feature selection we used a normalized depth scaling correlation threshold of −0.05, and total UMI count of more than 100 molecules. For metacell balanced k- nn graph construction we used K = 150. We computed 1,000 graph partitions resampling 75% of the cells to obtain a co-occurrence matrix. The final metacell partition was calculated on this co-occurrence matrix using K = 30, alpha=2 and min_mc_size=30 parameters. For 2D metacell projections, a force-directed projection was calculated using a K-nn constant of 20 and restricted the module graph degree by at most 4.

We filtered 21 metacells (representing 1,571 single cells) that lacked specific expression signatures (fewer than 50 genes with a fold-change ≥ 2) or that had a low number of total UMIs (fewer than 10,000 UMIs). We note that these filtered cells may represent weaker signals that are in fact part of other, more robust modules, but that for our goals in the analysis here, robustness of the reported transcriptional states and the subsequent genomic analyses is key. Overall, our final dataset consisted of 12,350 single-cells organized in 97 metacells. Gene expression fold enrichment and UMI counts were calculated for these metacells (Supplementary Data Files 1 and 2).

## Bacterial endosymbiont analysis

MARS-seq reads were mapped against the genomes of the two *Xenoturbella bocki* bacterial endosymbionts: related to *Chlamydia* and *Gammaproteobacteria*. The entire genome of each bacterium (in both frames) was considered for molecule counting, under the assumption that our polyA-capture based scRNA-seq method might be detecting at least some bacterial transcripts (even if inefficiently) and, therefore, these aggregated counts constitute a proxy for the presence of these symbionts within single-cells. Despite the fact that bacterial transcripts are non-polyadenlyated, we indeed detected hundreds of bacterial molecules (genome-wide) in some single-cells (0–4960 bacterial UMIs, median 10). Chlamydial and gammaproteobacterial UMI counts were pooled using our 97-metacell clustering solution and the signal for each bacteria was normalized by the total bacterial transcripts per cell and also the *Xenoturbella* transcripts (to account for differences in sampling efficiency).

## Functional gene annotation

We used blastp (with parameters −*evalue* 1e-5 and -*max_target_seqs* 1) to find the most similar, if any, human, fruit fly and yeast homologues (retrieved from Uniprot) for each protein of the *Xenoturbella* predicted proteomes. Additionally, we predicted, for each protein, the Pfam domain composition using Pfamscan[49] with default curated gathering threshold. TFs were identified using univocal Pfam domains for each structural TF family[50].

## Animal fixing and sectioning

Animals for in situ hybridisation were washed on ice progressively from sea-water into 3.6% MgCl$_2$ and anaesthetised in the dark at 4 °C for approximately four hours. Prior to complete loss of movement, the orientation of the animals was noted (anterior/posterior; left/right; dorsal/ventral axis) and individuals were photographed. Fully anaesthetised animals were placed on blotting paper on top of a block of frozen fixative (freshly prepared 4% paraformaldehyde (PFA) in 1× phosphate buffered saline (PBS)) and left for ~12 h in the dark at 4 °C to fix. Fixed animals were washed several times in 1× PBS and subsequently washed into progressively increasing concentrations of methanol (MeOH). Specimens were stored in 100% MeOH at −20 °C prior to use.

Fixed animals stored in 100% MeOH were removed from −20 °C, and MeOH was exchanged with 100% EtOH over a number of quick washes. Animals were washed into Histosol, three times for 20 min each at room temperature. After a total of one hour of Histosol incubation, samples were washed into a 50:50 mix of Histosol:paraffin wax. Two 30 min washes into the Histosol:paraffin mix were carried out at room temperature. Finally, specimens were transferred into 100% paraffin wax and incubated overnight at 60 °C. The following day, samples were washed five times in 100% paraffin wax at 60 °C. Paper moulds were filled with 100% molten wax and the specimens placed into the mould. Wax was left to solidify at room temperature.

Embedded animals were horizontally sectioned at either 6 or 7 μm onto Superfrost Plus slides along the dorso-ventral axis using a rotary microtome. Slides were stored at room temperature.

## Total RNA extraction and cDNA synthesis for ISH probe PCRs

Animals from which total RNA was to be extracted were placed directly into RNA-later (Sigma-Aldrich #R0109) for a minimum of one week for later RNA extraction. RNA was extracted using a standard Trizol/Chloroform RNA extraction protocol, quantified on Qubit and stored at −80 °C until required. Total RNA was used to make cDNA under RNAase-free conditions with the Ambion RETROscript RT kit (Invitrogen AM1710). A 12 μl reaction using 2 μl RNA and 1.2 μM oligo(dT) primers was incubated at 80 °C for three minutes and then removed onto ice. A final reaction volume of 20 μl was made with 1x M-MLV RT buffer, 10 mM dNTPs, 100U M-MLV reverse transcriptase and 10U RNase Inhibitor. Reverse transcription was carried out at 42 °C for two hours and the reaction finally heated to 90 °C for 10 min to inactivate the reverse transcriptase.

## Probe design and synthesis

cDNA from total RNA was diluted 1:10 for use in a PCR for probe synthesis PCR. PCR primer sequences for making in situ hybridisation probes are shown in Table S2. For each probe synthesis PCR, 20 μl first round reactions were prepared using Qiagen HotStarTaq DNA polymerase (Qiagen # 203205). The cycling conditions were as set as 95 °C for 15 min; 30 cycles at 94 °C for 30 s, 60 °C for 30 s, 72 °C for 1 min 30 s; a final elongation at 72 °C for 10 min. Resulting PCR products were diluted 1:50, and 1ul of the diluted reaction used in a second round PCR amplification, using reverse primers which also included a T7 promoter sequence. The cycling protocol was the same as that used in first round PCR. Second round PCR products were cleaned using Agencourt AMPure XP beads (Beckman Coulter #A63881) and visualised on a 1% agarose gel to verify amplicon size.

Antisense RNA probes were transcribed from the purified PCR products using T7 RNA polymerase with digoxigenin-conjugated dUTPs, with the Megascript T7 Transcription kit (Ambion, AM 1334). Final RNA probes were purified using RNA XP (Beckman, A63987) quantified on Qubit and stored at −80 °C in hybridization buffer.

## In situ hybridisation (ISH)

All preparatory and pre hybridisation stages were carried out in coplin jars treated with RNaseZap (Sigma Aldrich #R2020) and rinsed in DEPC-water. The protocol for in situ hybridisation on sections was adapted from Gillis et al.[51].

Slides were dewaxed by two five-minute washes in Histoclear, followed by two five-minute washes in 100% EtOH and then through two-minute washes through an ethanol gradient into diethylpyrocarbonate (DEPC)-treated phosphate-buffered saline (PBS) (90%, 70% and 50% EtOH in DEPC-PBS). Slides were rinsed once in DEPC-treated water; once in PBTw (0.1% Tween-20 in DEPC-PBS), and once in 2x SSC (0.3 M NaCl, 0.03 M sodium citrate).

Probes were diluted in 250 μl hybridisation solution (1× salt solution [0.2 M NaCl, 10 mM Tris pH 7.5, 5 mM NaH$_2$PO$_4$.H$_2$O, 5 mM Na$_2$HPO$_4$, 5 mM EDTA], 50% formamide, 10% dextran sulphate, 1 mg/ml

yeast tRNA, 1× Denhardt's solution) to a final concentration of 1 ng/µl. Hybridisation buffer containing probe was applied to each slide and incubated overnight at 60 °C under a glass coverslip in a sealed chamber humidified with 50% formamide/2× SSC. Slides were rinsed twice for 30 min each in wash solution at the hybridisation temperature, followed by three ten minute washes at room temperature in MABT (0.1 M maleic acid, 150 mM NaCl. 0.1% Tween-20, pH 7.5).

Slides were blocked for two hours at room temperature in blocking buffer composed of 1% Roche blocking reagent (Roche #11096176001) and 20% sheep serum, diluted in MABT. Anti-digoxigenin-AP Fab fragments (Roche #11093274910) were diluted 1:1000 in the same blocking buffer. 130 µl of the anti-DIG solution was applied to each slide and incubated overnight under a paraffin coverslip in a humidified chamber at room temperature.

Slides were rinsed twice in MABT, followed by five washes of progressive length in MABT on an orbital shaker, and equilibrated in NTMT (100 mM NaCl, 100 mM Tris, pH 9.5, 50 mM $MgCl_2$, 0.1% Tween-20). The colour reaction was initiated using BM Purple ready-to-use AP substrate (Roche #1142074001) in the dark at room temperature and stopped by washing in PBS. Slides were post-fixed in 4% PFA for 10 min, transferred back to PBS and mounted using Fluoromount G.

### Statistics and reproducibility

We sequenced the cellular transcriptomes and analysed cell types present in the four animal datasets. Statistical analysis and data visualization was performed in R unless mentioned otherwise. All in situ hybridizations were performed on at least 8 slides per probe. Representative animal sections were chosen for each gene. As very few animals are obtained from wild collections, animals were sectioned and multiple probes could then be tested on different sections taken from the same animal.

### Reporting summary

Further information on research design is available in the Nature Portfolio Reporting Summary linked to this article.

## Data availability

The single-cell RNA-seq data generated in this study and the transcriptome and genome sequences needed to reproduce the results are available in the publicly available GEO repository GSE211408. Source data are provided in this paper.

## Code availability

All code to generate the single-cell atlases, perform cross-species comparisons, and analyze gene modules is available on Github: https://github.com/sebepedroslab/Xenoturbella_sc_atlas. Unless otherwise specified, scripts are based on R version 3.5 and Python 3.7.10

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

## Acknowledgements

We thank Amos Tanay for initial input and feedback on data analyses. We also thank Kristineberg Center for Marine Research and Innovation for help in collecting samples. M.J.T., H.E.R., P.S. and A.-C.Z. were supported by ERC grant (ERC-2012-AdG 322790). Research in A.S.-P. group was supported by the European Research Council (ERC-StG 851647) and the Spanish Ministry of Science and Innovation (PID2021-124757NB-I00). X.G.-B. is supported by the European Union's H2020 research and innovation program under Marie Sklodowska-Curie grant agreement 101031767. Research in the H.M. group was supported by a Pilot Program Award from the Microbiome Center at the University of Chicago and by SESAME funding for the "Paris Single Cell Center".

## Author contributions

H.M. and M.J.T. conceived the study. H.M., M.J.T., and A.S.-P. designed and supervised the investigation. H.E.R., A.-C.Z., B.S., Y.L.-M., M.-P.M., P.S., and X.G.-B. performed experiments. H.E.R., A.S.-P., M.J.T., and H.M. wrote the paper. M.J.T. and H.M. supervised the investigation.

## Competing interests

The authors declare no competing interests.
