## [Peer Review File · Nature Communications]

Single cell atlas of *Xenoturbella bocki* highlights limited cell-type complexityREVIEWER COMMENTS

Reviewer #1 (Remarks to the Author):

Robertson et al describe the single cell atlas of the enigmatic species *Xenoturbella bocky* using single cell transcriptomics. They start with a few animals collected from the Fjord and dissociate their cells using a custom optimized protocol that uses enzymes in calcium magnesium free sea water. They then use FACS to sort live cells into plates, and amplify their transcripts using MARS-seq. The analysis uses Metacell, a package that partitions the dataset in small cell clusters that are likely to represent the same cell type. With this, they describe an abundance of gut cells in *Xenoturbella*, and provide transcriptomic details on the nervous system. They go on to analyze *Xenoturbella*'s muscle and pigmented cells and they derive an evolutionary interpretation, including a potential synapomorphy of the group *Xenoambulacraria* (the group itself is highly debated in the field). The dataset is of high quality, and compares with other datasets obtained previously with MARS-seq. Given the very interesting and controversial phylogenetic position of *Xenoturbella*, the dataset is also of key importance and will be key to understand more clues on this tricky evolutionary question.

The authors present very positive pieces of work, including the characterization of gut, neurons, muscle, a ciliary furrow cell type. The authors also go on and display the power of single cell transcriptomics in the analysis of symbiotic relationships, to describe the cell type specific signal of endosymbiotic bacteria. I think the paper addresses an important group of organisms, whose study is technically and operationally difficult. Thus, they provide a pioneering dataset that will likely go on to spur many other studies. I think the paper will be a strong candidate for publication in *Nature Communications* after revision.

The main concern on the present version is that it comes from a positioned optic that favors one of *Xenoturbella*'s highly contested phylogenetic affinities. The authors propose some potential synapomorphies. However, the single cell data that support these claims is thin, as presented in the current version. And the authors make no effort to show evidence for the alternative hypothesis. Surely, the authors are entitled to have and favor their own hypotheses. However, in the light of what is best for the article, the readership and the future of the debate, it would be better to present the data from a more pragmatic point of view. In fact, the data presented is a single cell dataset, and perhaps the evolutionary comparisons of cell types are beyond the scope of the current manuscript. If the authors wished to make solid claims on homology or at least resemblances (or "potential" homology as they call it) of the different cell types here the authors should perhaps perform more solid analyses, using integration tools or other cross-species analyses, with other species, including acoels, ambulacrarians but also spiralian. In my opinion, that would be beyond the scope of the current manuscript, as it is a very difficult task, using methods that have not matured yet, and would give material for a whole manuscript. The authors could go on and do that, or perhaps tone down some of the claims in the current manuscript with a more pragmatic approach. The authors should strengthen the evidence they present for the cell types they wish to make a claim, adding extra analyses.

One disappointment from reading the paper is that the authors do not attempt to identify stem cells. These have been described in the other species of the group, including several acoel species, by single cell methods and classic in situ approaches. While it is not taken for granted that *Xenoturbella* should have a population of piwi expressing stem cells similar to acoel neoblasts, it is a possibility worth contemplating, and the authors have the data to report it or at least report the absence of evidence. In fact, the authors leave some of their cells unclassified, and among which a "cyclin D2" is a marker. This uncharacterized fraction is distributed in different Metacells, so it is unclear which one expresses this. It would be good to address the question by plotting several known stem cell markers or at least, some proliferation markers.

Major comments:

-Excretory cell types. Perhaps this is one of the most critical pieces of information. The presence of an excretory cell type (protonephridia, metanephridia or nephrons) has been proposed as a synapomorphy of the "Nephrozoa". The presence of an excretory type in *Xenoturbella* then pretty much makes the reported absence of such type in acoels most likely a simplification. However, the authors present the data very thinly:

(Lines 99 - 106) "The largest group of metacells, defined principally by the expression of a number of transporter proteins"

Considering what the authors write in the next paragraph, it is key to address what transporters are these exactly. Here, it could suffice with a dotplot similar to that presented in figure 1H. But given that the claim on the excretory cell type is made on several transporters, it is key to consider which one is which. This is key as the excretory ("nephron like" function) has been suggested to be carried out by gut cells in acoels.

(Lines 107-112) "We also found a population of cells that expresses several genes - sodium transporters, an Rh-type ammonium transporter, and an ortholog of aquaporin-8 (Figure 1H) indicative of an osmoregulatory and excretory function. These aquaporin and ammonium transporter genes are specifically and highly expressed within this cell population and are not found expressed in the population of cells expressing transporters, suggesting the presence of cells specialised for excretion/osmoregulation and different from other gut cells".

But the authors only report a series of markers that are expressed in one metacell. Several of these markers can be considered bona fide markers of excretory types, but some others are not. There are aquaporins expressed in neurons, glia, glands, testes, ovaries, blood cell types and a range of other types (at least in mammals). This same concern applies to all genes reported in the figure. The authors could use supplemental figures to report dot plots similar to what they report in Figure 1H, but with other members of the families of genes reported in Figure 1H. Given that they have not managed to validate markers of this type, and that we are talking of one metacell, I think that more evidence is needed. This could be, with these markers, a ciliated neuron type, or a glial cell, for instance. Or, possibly a type that is different from gut at the transcriptome level, but embedded in the gut morphologically, consistent with what would be expected from previous reports.

-Muscle cell types: Here, I am not sure I follow the argument made by the authors. In the discussion the authors say (line 282) "representatives of any of the myogenic regulatory factors were found in our data", but it is unclear to me where in their figures this piece of information can be found. Perhaps the key to the argument (or my misunderstanding) is that the authors differentiate between striated and smooth mhc genes, but it is unclear to me a) if the separation between homologues is really that sharp that one can classify each unmistakably, b) how do they classify them and c) how this translates to their abstract conclusion ("We also suggest that Xenacoelomorpha muscles are likely to have evolved their "smooth" phenotype convergently"). Key to this would be to add more information on the difference between smooth and striated, including appropriate citations, as well as other analyses. Finally, the authors should clarify their conclusion in the text, as well as present alternatives.

-Pigment cells: Here, the authors make a point of a potential *Xenoambulacraria* synapomorphy because of the co-expression of *PKS* and *alx1* in sea urchin mesodermal cells and *Xenoturbella* pigment cell populations. But in order to check if this is a hypothesis worth contemplating, one would like to see where are these two transcripts expressed in other (non *Xenoambulacraria*) animals.

-Figure S2: this is an interesting (and pragmatic) analysis, but I do not find the text that refers to it in the main text, or a figure call. The authors might want to explain this in more detail and possibly migrate it to a full main figure.

Minor comments:

The name of the first author is misspelled in the manuscript file

Line 36: 2cm not small, compared to *C. elegans* or other worms.

Line 194: "In *Xenoturbella*, we also see the upregulation of *aristaless* (*alx1*) in the pigment cells (figure ref)" the figure call is missing – and the gene *alx1* does not appear in Figure 3A

Lines 378-382: The authors describe how they filtered 21 metacells (1571 cells) because they "lacked specific expression signatures". It would be good to know how they decided that they lack specific expression signatures.

Reviewer #2 (Remarks to the Author):

In NCOMMS-22-43703-T, "Single cell atlas of *Xenoturbella bocki* highlights the limited cell-type complexity of a non-vertebrate deuterostome lineage," Robertson, Sebe-Pedros and colleagues present a descriptive resource that aims to comprehensively describe adult cell types in a marine worm using gene expression data gleaned from whole-adult animal single cell RNA-Sequencing and RNA in situ hybridization. *Xenoturbella bocki* is a member of an enigmatic and understudied phylum, the Xenacoelomorpha, that may inform our understanding of the evolutionary origins of Bilaterian and/or Deuterostome body plans. The closest relatives of Xenacoelomorpha remain disputed, with recent work suggesting they are a sister clade to Ambulacrarians (hemichordates and echinoderms). While fleshing out a molecular and morphological understanding of *Xenoturbella bocki* anatomy is a worthwhile endeavor that contributes to this conversation, the new data provided here do not do much towards addressing the phylogenetic position of *X. bocki* and close relatives. As an outsider to the field – one who appreciates the unique challenges and opportunities posed by working with emerging research organisms, and the importance of sampling understudied taxa – I am left wanting more discussion of biological attributes that would make *X. bocki* and close relatives worthy of genetic and functional interrogation. These selling points would help to convey what members of the greater scientific community stand to benefit from the molecular anatomical characterization that has been presented here and would help justify why this work should appear in a journal with a broad audience, like Nature Communications.

I have a few comments and concerns with an eye towards "completeness" of the scRNA-Seq atlas. The data were acquired using MARS-Seq, a method that underperforms relative to other plate-based and droplet-based single cell and single nucleus RNA-Seq methods around gene detection (e.g., <https://www.nature.com/articles/s41587-020-0469-4#Sec8>). Choice of this technique may have depressed metrics (e.g., UMI and gene counts per cell). I understand that a lead author was involved in creation of the MetaCell algorithm used for analysis. I am curious about performance of Metacell relative to Seurat for creation of clustering groups/cell states and feel a general audience may benefit from discussion of what the benefits and drawbacks of the chosen methodology are relative to the stated goal of defining cell states and gold-standard biomarkers. Further exposition in figure legends (e.g., Figure 1C – what do the numbers on the nodes signify?) will assist readers that are used to seeing UMAPs and not metacell representations.

Additional points related to "completeness": 1) do *X. bocki* or other *Xenoturbella* species have cycling neoblast-like cells? Given that neoblasts have been described in acoelomorphs, I was surprised by the absence of neoblast-like clusters here. Given the deep conservation of neoblast markers across wide evolutionary distances and other conserved features of these cells I was surprised that the atlas

showed little obvious representation of cycling cells, neoblast-like cells, or the wandering germ cells reported in this species.

Comparisons of molecular and morphological similarity are made to hemichordates, and echinoderms, and in some cases to acoels. However, there are preprinted and published scRNA-Seq atlas data sets available for the acoels *I. pulchra* and *H. miamia* that are not grappled with here. Given the availability of homology-based approaches for inferring cell type/cell state identity in scRNA-Seq data, could this be a fruitful way of ascertaining what the “unknown” metacells are?

Points related to accessibility and utility of the data:

A. Your MAR-Seq data was aligned to an unpublished genome, yet transcriptome assemblies are publicly available (<https://academic.oup.com/gbe/article/10/9/2205/5068193>). Are there plans to make gene models and annotations available?

B. Please check figures (e.g., Figure 1F) and Table S2 to ensure that gene names, transcript/gene model numbers, and primary sequence for transcripts are available (in addition to primer sequences) so that others can easily ascertain whether they are working with the same transcript or the closest homolog.

REVIEWER COMMENTS

Reviewer #1 (Remarks to the Author):

Robertson et al describe the single cell atlas of the enigmatic species *Xenoturbella bocky* using single cell transcriptomics. They start with a few animals collected from the Fjord and dissociate their cells using a custom optimized protocol that uses enzymes in calcium magnesium free sea water. They then use FACS to sort live cells into plates, and amplify their transcripts using MARS-seq. The analysis uses Metacell, a package that partitions the dataset in small cell clusters that are likely to represent the same cell type. With this, they describe an abundance of gut cells in *Xenoturbella*, and provide transcriptomic details on the nervous system. They go on to analyze *Xenoturbella*'s muscle and pigmented cells and they derive an evolutionary interpretation, including a potential synapomorphy of the group *Xenoambulacraria* (the group itself is highly debated in the field). The dataset is of high quality, and compares with other datasets obtained previously with MARS-seq. Given the very interesting and controversial phylogenetic position of *Xenoturbella*, the dataset is also of key importance and will be key to understand more clues on this tricky evolutionary question.

The authors present very positive pieces of work, including the characterization of gut, neurons, muscle, a ciliary furrow cell type. The authors also go on and display the power of single cell transcriptomics in the analysis of symbiotic relationships, to describe the cell type specific signal of endosymbiotic bacteria. I think the paper addresses an important group of organisms, whose study is technically and operationally difficult. Thus, they provide a pioneering dataset that will likely go on to spur many other studies. I think the paper will be a strong candidate for publication in Nature Communications after revision.

We thank the reviewer for the kind comments on the manuscript. We are also grateful for the constructive comments and found that addressing these have increased the clarity, completeness and overall value of the manuscript. Please find our responses to specific comments below as well as references to associated changes in the manuscript.

The main concern on the present version is that it comes from a positioned optic that favors one of *Xenoturbella*'s highly contested phylogenetic affinities.

We appreciate this reviewer's perspective, as indeed, the position of *Xenoturbella* is not definitively settled. We have revised the text to reflect a more neutral position which allows for the two most widely held views regarding potential positions of *Xenoturbella* 1) as sister to Bilateria or 2) as sister to Ambulacraria.

The authors propose some potential synapomorphies. However, the single cell data that support these claims is thin, as presented in the current version. And the authors make no effort to show evidence for the alternative hypothesis. Surely, the authors are entitled to have and favor their own hypotheses. However, in the light of what is best for the article, the readership and the future of the debate, it would be better to present the data from a more pragmatic point of view. In fact, the data presented is a single cell dataset, and perhaps the evolutionary comparisons of cell types are beyond the scope of the current manuscript. If the authors wished to make solid claims on homology or at least resemblances (or "potential" homology as they call it) of the different cell types here the authors should perhaps perform more solid analyses, using integration tools or other cross-species analyses, with other species, including acoels, ambulacrarians but also spiralian. In my opinion, that would be beyond the scope of the current manuscript, as it is a very difficult task, using methods that have not matured yet, and would give material for a whole manuscript. The authors could go on and do that, or perhaps tone down some of the claims in the current manuscript with a more pragmatic approach. The authors should strengthen the evidence they present for the cell types they wish to make a claim, adding extra analyses.

We have revised our suggestions of cell type homologies to be more conservative and to allow for alternative interpretations. Specific examples include:

- 1) Discussion of excretory cell types has been revised**
- 2) Discussion of muscle cell types has been revised based on identification of cycling cell markers in cluster 76**
- 3) Discussion of pigment cell type homology has been revised**

One disappointment from reading the paper is that the authors do not attempt to identify stem cells. These have been described in the other species of the group, including several acoel species, by single cell methods and classic in situ approaches. While it is not taken for granted that *Xenoturbella* should have a population of piwi expressing stem cells similar to acoel neoblasts, it is a possibility worth contemplating, and the authors have the data to report it or at least report the absence of evidence. In fact, the authors leave some of their cells unclassified, and among which a “cyclin D2” is a marker. This uncharacterized fraction is distributed in different Metacells, so it is unclear which one expresses this. It would be good to address the question by plotting several known stem cell markers or at least, some proliferation markers.

We share this reviewer’s interest in putative stem cell populations and were intrigued by their astute observation of cyclin D2 in metacell “unknown 9”. This prompted us to perform a cross-species analysis utilizing the published *Hofstenia miamia*, *Isodiametra pulchra* and *Nematostella vectensis* datasets. Intriguingly, we identified commonly associated “stem” cell populations in pairwise comparisons between *Xenoturbella* and all three of the compared lineages. In all cases, similarities were centered on “unknown 9”. We further examined the list of genes found to be driving these associations and have reported these in the text.

Major comments:

-Excretory cell types. Perhaps this is one of the most critical pieces of information. The presence of an excretory cell type (protonephridia, metanephridia or nephrons) has been proposed as a synapomorphy of the “Nephrozoa”. The presence of an excretory type in *Xenoturbella* then pretty much makes the reported absence of such type in acoels most likely a simplification. However, the authors present the data very thinly:

(Lines 99 - 106) “The largest group of metacells, defined principally by the expression of a number of transporter proteins”

Considering what the authors write in the next paragraph, it is key to address what transporters are these exactly. Here, it could suffice with a dotplot similar to that presented in figure 1H. But given that the claim on the excretory cell type is made on several transporters, it is key to consider which one is which. This is key as the excretory (“nephron like” function) has been suggested to be carried out by gut cells in acoels.

(Lines 107-112) “We also found a population of cells that expresses several genes - sodium transporters, an Rh-type ammonium transporter, and an ortholog of aquaporin-8 (Figure 1H) indicative of an osmoregulatory and excretory function. These aquaporin and ammonium transporter genes are specifically and highly expressed within this cell population and are not found expressed in the population of cells expressing transporters, suggesting the presence of cells specialised for excretion/osmoregulation and different from other gut cells”.

But the authors only report a series of markers that are expressed in one metacell. Several of these markers can be considered bona fide markers of excretory types, but some others are not. There are aquaporins expressed in neurons, glia, glands, testes, ovaries, blood cell types and a range of other types (at least in mammals). This same concern applies to all genes reported in the figure. The authors could use supplemental figures to report dot plots similar to what they report in Figure 1H, but with other members of the families of genes reported in Figure 1H. Given that they have not managed to validate markers of this type, and that we are talking of one metacell, I think that more evidence is needed. This could be, with these markers, a ciliated neuron type, or a glial cell, for instance. Or, possibly a type that is different from gut at the transcriptome level, but embedded in the gut morphologically, consistent with what would be expected from previous reports.

We have revised the labeling of these cells in Figure 1 to remove the “excretory cell” label. In addition, we now refer to these cells as an unknown population in the figure legend for Figure 1. In order to more accurately assess transcriptional similarities for these cells, we performed cross-species analyses for cell type similarities which is now included as Supplemental Figure 5.

We feel that the intriguing results of the cross-species analysis (Supplemental Figure 5) will warrant additional study in the future and provides an exciting new direction for additional work to delve into the spatial expression of many of these genes as well as the orthology of the transporters. Accordingly, we have used a more cautious approach in our assignment of an identity to these cells and have revised our discussion around these cell types.

-Muscle cell types: Here, I am not sure I follow the argument made by the authors. In the discussion the authors say (line 282) “representatives of any of the myogenic regulatory factors were found in our data”, but it is unclear to me where in their figures this piece of information can be found. Perhaps the key to the argument (or my misunderstanding) is that the authors differentiate between striated and smooth mhc genes, but it is unclear to me a) if the separation between homologues is really that sharp that one can classify each unmistakably, b) how do they classify them and c) how this translates to their abstract conclusion (“We also suggest that Xenacoelomorpha muscles are likely to have evolved their “smooth” phenotype convergently”). Key to this would be to add more information on the difference between smooth and striated, including appropriate citations, as well as other analyses. Finally, the authors should clarify their conclusion in the text, as well as present alternatives.

In the process of examining putative stem cell relationships, we also generated a mapping of a list of cyclin cell markers. Interestingly, one of our muscle clusters, cluster 76, was found to be a cycling population or a population which has recently emerged from a cycling cell population. In our previous description of this population in the text, we had noted that this cluster was transcriptionally distinct from the other 7 muscle clusters, clusters 69-75. Given our new knowledge that cluster X represents a cycling/recently cycling population, it is more plausible that the unique expression of transcription factors and muscle genes here are those expressed in early differentiating muscle. We have revised the text to reflect this finding and thank this reviewer for their careful assessment of the muscle features discussed in the manuscript.

-Pigment cells: Here, the authors make a point of a potential Xenoambulacraria synapomorphy because of the co-expression of PKS and alx1 in sea urchin mesodermal cells and *Xenoturbella* pigment cell populations. But in order to check if this is a hypothesis worth contemplating, one would like to see where are these two transcripts expressed in other (non Xenoambulacrarian) animals.

In order to more formally compare the PKS/alx+ cells in *Xenoturbella* to sea urchins, we performed an additional SAMAP comparison between sea urchins and *Xenoturbella*. Despite the presence of PKS and alx in *Xenoturbella* cells, reminiscent of sea urchin mesenchyme, there was no support in this analysis to indicate a shared homology of *Xenoturbella* pigmented cells with sea urchin mesenchyme cells.

-Figure S2: this is an interesting (and pragmatic) analysis, but I do not find the text that refers to it in the main text, or a figure call. The authors might want to explain this in more detail and possibly migrate it to a full main figure.

Figure S2 is now referenced in the text. Additional comment on Figure S2 is now found in the results and discussion section.

Minor comments:

The name of the first author is misspelled in the manuscript file

We thank this reviewer for pointing out this error. We have corrected the misspelling.

Line 36: 2cm not small, compared to *C. elegans* or other worms.

We have updated the text to remove mention of *Xenoturbella*'s “small” size. As the reviewer rightly points out, there are many examples of smaller worms.

Line 194: “In *Xenoturbella*, we also see the upregulation of aristaless (alx1) in the pigment cells (figure ref)” the figure call is missing – and the gene alx1 does not appear in Figure 3A

Yes, alx1 (labeled alx) is present in figure 3A dotmap. We have highlighted on a larger version of the panel below.

Lines 378-382: The authors describe how they filtered 21 metacells (1571 cells) because they “lacked specific expression signatures”. It would be good to know how they decided that they lack specific expression signatures.

We now indicate in the methods section the criteria for filtering these metacells (low number of specific genes and/or low total number of UMIs).

Reviewer #2 (Remarks to the Author):

In NCOMMS-22-43703-T, “Single cell atlas of *Xenoturbella bocki* highlights the limited cell-type complexity of a non-vertebrate deuterostome lineage,” Robertson, Sebe-Pedros and colleagues present a descriptive resource that aims to comprehensively describe adult cell types in a marine worm using gene expression data gleaned from whole-adult animal single cell RNA-Sequencing and RNA in situ hybridization. *Xenoturbella bocki* is a member of an enigmatic and understudied phylum, the Xenacoelomorpha, that may inform our understanding of the evolutionary origins of Bilateral and/or Deuterostome body plans. The closest relatives of Xenacoelomorpha remain disputed, with recent work suggesting they are a sister clade to Ambulacrarians (hemichordates and echinoderms). While fleshing out a molecular and morphological understanding of *Xenoturbella bocki* anatomy is a worthwhile endeavor that contributes to this conversation, the new data provided here do not do much towards addressing the phylogenetic position of *X. bocki* and close relatives. As an outsider to the field – one who appreciates the unique challenges and opportunities posed by working with emerging research organisms, and the importance of sampling understudied taxa – I am left wanting more discussion of biological attributes that would make *X. bocki* and close relatives worthy of genetic and functional interrogation. These selling points would help to convey what members of the greater scientific community stand to benefit from the molecular anatomical characterization that has been presented here and would help justify why this work should appear in a journal with a broad audience, like Nature Communications.

We thank this reviewer for their insightful comments. We have revised with a broader audience in mind and have additionally included a comparative analysis that will further address the importance of this study to our understanding of animal diversity more broadly.

The text has been modified to more clearly draw attention to the reasoning behind the characterization of this species using scRNA-seq. We also think that our new SAMPap analysis provides exciting support for shared features of some cell types and further highlights the exciting potential of evolutionary studies of cell diversity. Notably, and rather unexpectedly, this analysis uncovered similarities between a previously unannotated *Xenoturbella* cell type and stem cell-like populations in acoels and cnidarians. While it remains to be determined whether these features arise by convergence or homology, we feel that the identification of molecular similarities will provide exciting new directions for researchers examining the origins of stemness in animals.

I have a few comments and concerns with an eye towards “completeness” of the scRNA-Seq atlas. The data were acquired using MARS-Seq, a method that underperforms relative to other plate-based and droplet-based single cell and single nucleus RNA-Seq methods around gene detection (e.g., <https://www.nature.com/articles/s41587-020-0469-4#Sec8>). Choice of this technique may have depressed metrics (e.g., UMI and gene counts per cell). I understand that a lead author was involved in creation of the MetaCell algorithm used for analysis. I am curious about performance of Metacell relative to Seurat for creation of clustering groups/cell states and feel a general audience may benefit from discussion of what the benefits and drawbacks of the chosen methodology are relative to the stated goal of defining cell states and gold-standard biomarkers. Further exposition in figure legends (e.g., Figure 1C – what do the numbers on the nodes signify?) will assist readers that are used to seeing UMAPs and not metacell representations.

Like Metacell, Seurat employs a kNN similarity graph to represent cell-cell transcriptional similarity. An important difference is the algorithm later used to cluster cells based on this graph. Seurat offers several community detection algorithms (e.g. Leiden or Louvain), while Metacell performs more aggressive clustering to generate groups of cells that are highly similar/cohesive, but without necessarily maximizing distinctiveness between clusters. More recent implementations of the idea of metacells (namely, SEACells <https://www.nature.com/articles/s41587-023-01716-9>) retrieve an arbitrary, user-defined number of (highly-redundant) metacells. The full details on these differences, as well as the general principles motivating the metacell approach, are fully developed in the original Metacell publication and we believe that revisiting this discussion here is out of the scope of the paper.

Still, to address the reviewer’s concerns, we have compared Seurat-based clustering with our metacell clusters. A major difference can result from the usage of different methods to retrieve variable marker genes for clustering. So we tested Seurat with each of the variable marker selections methods it offers, and also feeding it with the markers we originally used for the metacell clustering. This UpSet plot shows the distinctiveness of these marker sets:

And, yet, despite these differences, the cell clusters obtained in Seurat strongly resembled some of our metacells. Here's an example comparison (cluster-to-cluster correlation, based on top2000 variable genes) of metacells (y-axis) versus Seurat clusters (x-axis):

Seurat clustering with the 3 other variable marker gene sets resulted in similar results. The main differences (defined as a metacell with no Seurat cluster with a gene expression correlation > 0.5) found in all cases were the absence of *muscle-like*, *gland 2*, and *unknown 3* cell types in Seurat clustering. In the reciprocal analysis, no Seurat clusters were found to be absent/not represented by metacells.

The overall conclusion is that cell type clusters are stable to different algorithms, which is not entirely surprising given the strong, cell type-specific gene expression signatures (hundreds of genes being cell type-specific) that make the approaches robust even to highly distinct marker gene sets.

Additional points related to “completeness”: 1) do *X. bocki* or other *Xenoturbella* species have cycling neoblast-like cells? Given that neoblasts have been described in acoelomorphs, I was surprised by the absence of neoblast-like clusters here. Given the deep conservation of neoblast markers across wide evolutionary distances and other conserved features of these cells I was surprised that the atlas showed little obvious representation of cycling cells, neoblast-like cells, or the wandering germ cells reported in this species.

This is a very valid question. It should be noted that both reviewers raised this point. To address this question, we performed cell type comparisons using a published computational pipeline, SAMAP. We compared published *Hofstenia miamia*, *Isodiametra pulchra* and *Nematostella vectensis* datasets to the

***Xenoturbella* data. Intriguingly, we identified commonly associated “stem” cell populations in pairwise comparisons between *Xenoturbella* and all three of the compared lineages. In all cases, similarities were centered on “unknown 9”. Specific genes found to drive this association have been indicated in the text. We find the results of this analysis extremely exciting as they could indicate some shared molecular features of stemness and will provide myriad future directions for investigation into these stem programs.**

Comparisons of molecular and morphological similarity are made to hemichordates, and echinoderms, and in some cases to acoels. However, there are preprinted and published scRNA-Seq atlas data sets available for the acoels *I. pulchra* and *H. miamia* that are not grappled with here. Given the availability of homology-based approaches for inferring cell type/cell state identity in scRNA-Seq data, could this be a fruitful way of ascertaining what the “unknown” metacells are?

Please see the our response to Review 1 above. To address potential homology or similarity of some of the unknown cells, we have performed a SAMap analysis using a previously published analysis tool for this purpose (Tarashansky et al, 2021, ELIFE)

Points related to accessibility and utility of the data:

A. Your MAR-Seq data was aligned to an unpublished genome, yet transcriptome assemblies are publicly available (<https://academic.oup.com/gbe/article/10/9/2205/5068193>). Are there plans to make gene models and annotations available?

A publication, in press, from the Telford lab is the source of the genome in this study. This genome as well as models we generated and utilized for assigned reads to genes are included in the data repository for this paper which can be found at (GSE211408). And the token for reviewer access anodoouifzevud

B. Please check figures (e.g., Figure 1F) and Table S2 to ensure that gene names, transcript/gene model numbers, and primary sequence for transcripts are available (in addition to primer sequences) so that others can easily ascertain whether they are working with the same transcript or the closest homolog.

We appreciate the importance of providing complete resource data and supporting orthology assignment information. In addition to the genome and gene models that were utilized for mapping and assignment of reads to genes, we have also deposited the gene annotations so that our naming of genes is readily available (GSE211408). And the token for reviewer access anodoouifzevud

REVIEWERS' COMMENTS

Reviewer #1 (Remarks to the Author):

The authors have satisfactorily answered my queries. I think that the cell type atlas of *Xenoturbella* is an exciting manuscript that will be fundamental for our understanding of the evolution of cell types, as well as for further elucidating the phylogenetic position of the Xenacoelomorpha group.

The authors could have considered changing the title to accommodate the two contentious views of the groups phylogenetic position instead of favouring one. However, I think that these are now fairly explained in the text - it is obvious that the authors, which have contributed significantly to one of these phylogenetic views, are in favour of it. Readers are presented both in the text and can judge each of them with this paper and previous literature.

Reviewer #2 (Remarks to the Author):

Dr Marlow and colleagues provide a well documented single cell transcriptomic resource for *Xenoturbella bocki*, a member of the Xenacoelomorpha phylum. The MARS-Seq analysis is clearly presented and the in vivo ISH validation is well-explained in the context of *X. bocki* anatomy. The manuscript has been strengthened considerably by additional efforts undertaken by the authors to perform cross-species orthology based comparisons of their metacell data with 2 published acoel (*H. miamia* and *I. pulchra*), sea urchin, and *Nematostella* datasets using SAMap. These comparisons provide further corroborative evidence for well-characterized and abundant differentiated cell types highlighted in the atlas, and provide points of discussion regarding shared or divergent properties for major anatomical systems (e.g., the nervous system, musculature, and pigment system). Intriguingly, unknown metacell(s) (e.g., Unknown 9) showed putative homology to acoel and *nematostella* adult pluripotent stem cell populations, an observation that may fuel inquiries into tissue homeostasis and cell fate specification in the future.

The authors have addressed reviewer concerns adequately and have strengthened the reporting of their resource. I support acceptance and publication of the manuscript in Nature Communications.

REVIEWERS' COMMENTS

Reviewer #1 (Remarks to the Author):

The authors have satisfactorily answered my queries. I think that the cell type atlas of *Xenoturbella* is an exciting manuscript that will be fundamental for our understanding of the evolution of cell types, as well as for further elucidating the phylogenetic position of the Xenacoelomorpha group.

The authors could have considered changing the title to accommodate the two contentious views of the groups phylogenetic position instead of favouring one. However, I think that these are now fairly explained in the text - it is obvious that the authors, which have contributed significantly to one of these phylogenetic views, are in favour of it. Readers are presented both in the text and can judge each of them with this paper and previous literature.

Our Response: In shortening the title to meet Nature Communications formatting requirements, we have also addressed this reviewer's remaining concern with regard to accommodating the two views of the phylogenetic position of *Xenoturbella*. We agree that this title is more reflective of our findings and thank the reviewer for their suggestion.

Reviewer #2 (Remarks to the Author):

Dr Marlow and colleagues provide a well documented single cell transcriptomic resource for *Xenoturbella bocki*, a member of the Xenacoelomorpha phylum. The MARS-Seq analysis is clearly presented and the in vivo ISH validation is well-explained in the context of *X. bocki* anatomy. The manuscript has been strengthened considerably by additional efforts undertaken by the authors to perform cross-species orthology based comparisons of their metacell data with 2 published acoel (*H. miamia* and *I. pulchra*), sea urchin, and *Nematostella* datasets using SAMap. These comparisons provide further corroborative evidence for well-characterized and abundant differentiated cell types highlighted in the atlas, and provide points of discussion regarding shared or divergent properties for major anatomical systems (e.g., the nervous system, musculature, and pigment system). Intriguingly, unknown metacell(s) (e.g., Unknown 9) showed putative homology to acoel and nematostella adult pluripotent stem cell populations, an observation that may fuel inquiries into tissue homeostasis and cell fate specification in the future.

The authors have addressed reviewer concerns adequately and have strengthened the reporting of their resource. I support acceptance and publication of the manuscript in Nature Communications.

Our Response: We thank the reviewer for their comments.